# Cell-state transitions and collective cell movement generate an endoderm-like region in gastruloids

Ali Hashmi[1], Sham Tlili[1], Pierre Perrin[1], Molly Lowndes[2,3], Hanna Peradziryi[2], Joshua M Brickman[2,3], Alfonso Martínez Arias[4,5], Pierre-François Lenne[1]*

[1]Aix-Marseille University, CNRS, UMR 7288, IBDM, Turing Center for Living Systems, Marseille, France; [2]Novo Nordisk Foundation Center for Stem Cell Biology (DanStem), University of Copenhagen, Copenhagen, Denmark; [3]Novo Nordisk Foundation Center for Stem Cell Medicine (reNEW), University of Copenhagen, Copenhagen, Denmark; [4]Department of Genetics, University of Cambridge, Cambridge, United Kingdom; [5]Systems Bioengineering, DCEXS , Universidad Pompeu Fabra, ICREA, Barcelona, Spain

*For correspondence:
pierre-francois.lenne@univ-amu.fr

Competing interest: The authors declare that no competing interests exist.

**Abstract** Shaping the animal body plan is a complex process that involves the spatial organization and patterning of the different germ layers. Recent advances in live imaging have started to unravel the cellular choreography underlying this process in mammals, however, the sequence of events transforming an unpatterned cell ensemble into structured territories is largely unknown. Here, using gastruloids –3D aggregates of mouse embryonic stem cells- we study the formation of one of the three germ layers, the endoderm. We show that the endoderm is generated from an epiblast-like homogeneous state by a three-step mechanism: (i) a loss of E-cadherin mediated contacts in parts of the aggregate leading to the appearance of islands of E-cadherin expressing cells surrounded by cells devoid of E-cadherin, (ii) a separation of these two populations with islands of E-cadherin expressing cells flowing toward the aggregate tip, and (iii) their differentiation into an endoderm population. During the flow, the islands of E-cadherin expressing cells are surrounded by cells expressing T-Brachyury, reminiscent of the process occurring at the primitive streak. Consistent with recent in vivo observations, the endoderm formation in the gastruloids does not require an epithelial-to-mesenchymal transition, but rather a maintenance of an epithelial state for a subset of cells coupled with fragmentation of E-cadherin contacts in the vicinity, and a sorting process. Our data emphasize the role of signaling and tissue flows in the establishment of the body plan.

## Editor's evaluation

The authors took advantage of the gastruloid system to explore mechanisms of endoderm specification at cellular resolution. First, they confirm that contrary to mesoderm, nascent endoderm does not undergo epithelial-mesenchymal transition. Second, they provide evidence for a role of tissue flow, and possibly heterogeneity of cellular junction tension, in the sorting and differentiation of the islets of endoderm cells.

## Introduction

Gastrulation is a crucial stage of animal development that results in the laying down of the body plan through the specification of the germ layers – the endoderm, the mesoderm, and the ectoderm – and their organization with reference to three orthogonal axes (*Wolpert et al., 2006*; *Solnica-Krezel and*

*Sepich, 2012*; *Williams and Solnica-Krezel, 2017*). Defects in gastrulation result in premature death or birth defects (*Ferrer-Vaquer and Hadjantonakis, 2013*). In most vertebrates, gastrulation entails the reorganization of a large mass of cells that has been generated in a burst of cell divisions and the process can be easily followed and manipulated in the lab as the embryos develop externally. Mammalian embryos are different, in that an increase in the cellular mass is concomitant with gastrulation (*Snow, 1977*; the fact that they develop in the uterus of the mother makes this coupling and the process of gastrulation difficult to study). For this reason, our understanding of mammalian gastrulation is poor relying on mutant analysis and description of patterns of gene expression (*Anderson and Ingham, 2003*; *Artzt, 2012*). Recent live imaging of the process (*McDole et al., 2018*) has added an important dimension to our understanding of the event, but this is low throughput and involves time consuming analysis. Gastrulation is, essentially, a cellular process that needs to be investigated at the cellular level and it would be useful to complement the in vivo studies with alternatives in vitro.

The advent of in vitro models of mammalian development provides several advantages (*Fu et al., 2021*). Firstly, unlike embryos, such systems are amenable to systematic perturbations. The system size and its composition can be conveniently modulated by changing the type and the initial number of cells seeded. Secondly, environmental cues – not limited to morphogen concentration (*Heemskerk et al., 2019*), such as geometric confinements and properties of extracellular matrix – can be finely tuned. Lastly, such systems are amenable to live imaging at single cell resolution to follow cell fates and cell dynamics.

Gastruloids offer the opportunity to enrich our understanding of the agents that may influence gastrulation/embryogenesis – particularly the role of environment, forces, and signaling. Upon exposure to external cues and morphogens, gastruloids show an intrinsic capability to polarize, elongate and pattern in a manner that is reminiscent of the early embryo (*van den Brink et al., 2014*; *Turner et al., 2017*). Strikingly, 3D gastruloids exhibit a multiaxial organization and spatiotemporal patterning of Hox genes (*Beccari et al., 2018*) with signatures of somitogenesis (*van den Brink et al., 2020*; *Veenvliet et al., 2020*). From in vitro assays, it was shown that combinations of embryonic and extraembryonic stem cells is needed to generate a pro-amniotic cavity (*Harrison et al., 2017*) and gastrulating embryo-like structures (*Sozen et al., 2018*). Moreover, by exploiting aggregates, the role of mechanics – for instance a relationship between the physical contact of cells with their confinement and the expression of the mesoderm transcription factor Brachyury (T-Bra) – has been revealed (*Sagy et al., 2019*). Besides tridimensional (3D) systems, planar/2D gastruloids confined on micropatterns have emphasized the role of the boundary conditions, shapes (*Blin et al., 2018*) and scaling parameter on the patterning of the different germ layers (*Warmflash et al., 2014*; *Etoc et al., 2016*; *Morgani et al., 2018*; *Xue et al., 2018*).

However, how a genetically unpatterned and morphologically symmetric 3D ESCs aggregate transforms into a spatially organized and patterned array of germ layers is largely unknown. In particular, how cellular movements, their rearrangements and patterning coordinate has not been explored. In embryo, the widespread notion is that the endoderm is formed from a single pool of cells that firstly become mesenchymal, then migrate and a proportion of which eventually re-epithelialize (*Williams et al., 2012*). This notion has been recently challenged by in vivo observations that show that the definitive endoderm is formed in absence of an epithelial to mesenchymal transition (EMT) (*Scheibner et al., 2021*).

Here, we dissect the formation of an endoderm-like region in 3D gastruloids, focusing on the spatio-temporal deployment and cell state transitions that yield the aforesaid region; specifically, starting from an ensemble of epithelial cells, we follow the cell-state transitions that govern the mesendodermal-like progenitors by looking at expression of the transcription factor Brachyury (T-Bra) and E-cadherin (E-cad). We find that an endoderm-like region is established from a distinct pool of cells, different from the mesoderm. Furthermore, we identify tissue scale flow that localizes the progenitors of the endoderm at a pole. Lastly, we find that a heterogeneity of cellular junction tension could be responsible for segregating the endoderm-like region from the rest of the aggregate via a cell-sorting mechanism.

## Results

### A pole of E-cadherin expression emerges at the tip of the aggregate

To induce a pluripotent, post-implantation epiblast state of the mouse ESCs (*Hayashi et al., 2011*), we placed the cells in Activin and FGF throughout the protocol (*Figure 1A*). A portion of the aggregates were exposed to the Wnt agonist Chiron (Chi) as the combination with Activin and Chi tends to promote endoderm formation (*Figure 1A*; *Morgani et al., 2018*; *Engert et al., 2013*). We investigated the aggregate morphologies and gene expression patterns following a pulse of Chi (lasting a day from the second day post plating).

Aggregates exposed to Chi lose their spherical morphology and acquire a teardrop shape (*Figure 1B*, day 4, *Video 1*). We find that 30/37 or ~80% aggregates qualify as elongated with an elongation threshold set to 0.125 (see also *Video 2*). In contrast, 15/20 or 75% of control aggregates remained spherical (*Figure 1b*, day 4). At the pointed end of the Chi exposed aggregates, we observed an E-cadherin (E-cad)-rich region, which was absent in the control aggregates (*Figure 1C*, *Figure 1—figure supplement 1*). The E-cad pole emerges from an initially homogeneous distribution of E-cad (*Figure 1D*, *Video 3*). Likewise, a T-Brachyury (T-Bra)-rich pole, clearly visible at the tip of the aggregate at day 4, emerges from a spatially homogeneous distribution at day 3.

Time-lapse imaging shows distinct time evolution of global E-cad and T-Bra signals. While E-cad is expressed from the early time points (*Figure 1D*, day 2), T-Bra expression follows exposure to Chi (*Figure 1D* and *Video 4*). The Chi pulse triggers a decay in E-cad level (mean intensity, *Figure 1E* and *Video 5*). T-Bra level, maximal just before removal of Chi, gradually decreases thereafter (*Videos 2 and 6*).

Most aggregates show a single pole of T-Bra (35/64), some display multiple poles (18/64) and 11/64 do not polarize (*Video 2*). We then asked whether the polarized expression of E-cad precedes, follows or is concomitant with the deformation of the aggregates. Using descriptors of aggregate's shape and polarization of fluorescence signal (see Materials and methods), we found that E-cad and T-Bra polarize prior to the onset of tip formation (*Figure 1F and G*), suggesting a possible role of E-cad and T-Bra in the aggregate's shape polarization.

### E-cadherin expression defines an endoderm-like region in the gastruloid

To resolve the spatial distribution of E-cad and T-Bra expressing cells at the single-cell level, aggregates exposed to Chi were stained at day 4. As shown in *Figure 2A*, a group of E-cad expressing cells is localized at the tip, spatially segregated from, and surrounded by T-Bra expressing cells. Furthermore, we observed that this core expresses Sox17 (*Figure 2A*, inset, *Figure 2—figure supplement 1*) and Foxa2 (*Figure 2—figure supplement 1*, *Video 7*) – markers of the endoderm. We confirmed that Sox2 is present earlier but absent at 5 days-pp while Sox17 is retained (*Figure 2—figure supplement 1*). These observations lead us to conclude that the group of E-cad expressing cells demarcates an endoderm-like region in the gastruloids. *Figure 2B* shows a histogram of the various cell populations at the tip of aggregates. The core of E-cad/Sox17 expressing cells and the T-Bra expressing cells constitute the majority of the cell population near the tip (*Figure 2B*). Nevertheless, a minority outside the core expresses either Sox17 alone or in conjunction with T-Bra. To obtain some insight into the supracellular tissue organization, we further determined the local neighborhood of the different cell populations (*Figure 2C*). The E-cad/Sox17 expressing cells that form the core maintain a relatively larger simply-connected-component (126 of 131 Ecad+/Sox17+ cells) and a homotypic neighborhood (111 E-cad+/Sox17+ cells maintain an Ecad+/Sox17+majority neighborhood), which implies that they form a tightly packed group of cells. In contrast, the T-Bra expressing cells form multiple simply connected-components (20 components) suggesting that they are sparsely distributed around the core. The E-cad+/Sox17+ cells thus form a compact region that is flanked by sparse T-Bra+ cells and few other cell types that span widely about the tip (*Figure 2D* and *Figure 2—figure supplement 2* for two other aggregates).

### Endoderm is established from an epiblast-like population

To probe how the endoderm-like region is established in gastruloids, we performed time-lapse imaging of the aggregates at single cell resolution. At the early stages E-cad expression pervades the entire

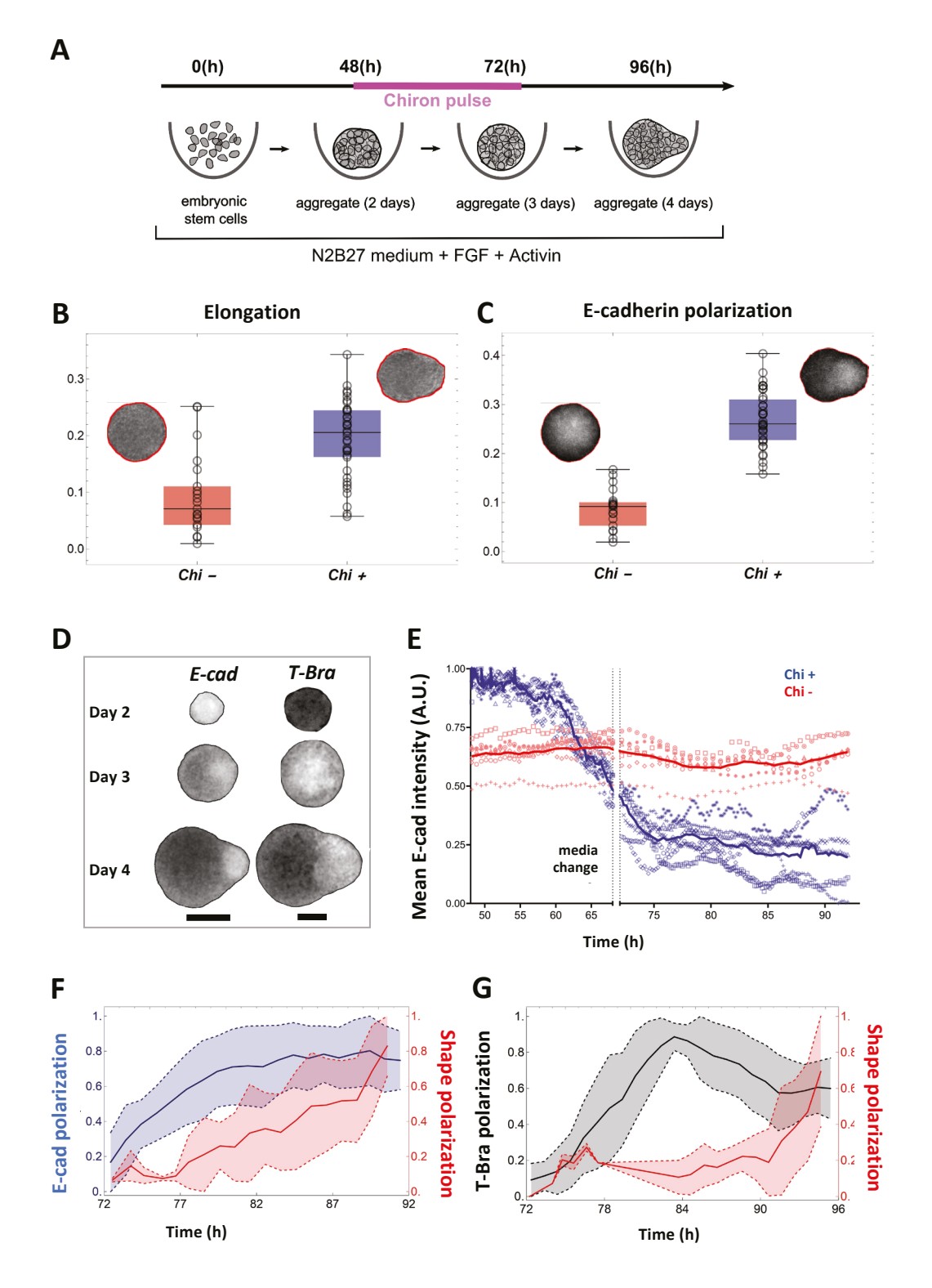

**Figure 1.** Signal and shape polarization of gastruloids. (**A**) Schematic of the experimental protocol; mESCs seeded onto a low-adherence substrate form aggregates that are exposed to N2B27 medium, FGF and Act throughout the experiment. In addition, a pulse of Chi is provided for 24 hr - starting at 48 hr-post plating. (**B**) The box-whisker plot shows the distribution of elongation for Chi + (n = 37) and the control (Chi -) aggregates (n = 20) (Mann-Whitney test with correction, p-value = 3 x 10⁻⁵, N = 2 replicates). The insets delineate the contours of the two classes of aggregates. (**C**) Box-whisker plot showing the distribution of E-cad polarization in the Chi + (n = 37) and Chi - aggregates (n = 20). The polarization of the signal is defined by the

*Figure 1 continued on next page*

*Figure 1 continued*

maximum contrast between the two halves of the aggregate (Mann-Whitney test with correction, p = 7 x 10⁻¹⁰, N = 2 replicates). The insets display E-cad intensity within the aggregates. (**D**) Snapshots of aggregates (first column composed of E-cad-GFP/Oct4-mCherry and the second column composed of T-Bra-GFP/NE-mKate2 and E14Tg2a.4) exposed to Chi pulse at different developmental stages (scale bars for the columns represent 100 µm). The T-Bra-GFP/NE-mKate2 aggregate was generated from ~200 cells. (**E**) Temporal changes in the mean E-cad intensity profile between 48 and 92 hr for aggregates (Ecad-GFP/Oct4-mCherry) exposed to Chi (n = 7 aggregates, N = 3 replicates) and the control (n = 7 aggregates, N = 1 replicate). (**F**) The plot illustrates the polarization of E-cad (n = 14 aggregates, N = 3 replicates) with respect to the deformation of the aggregates between 72 and 92 hr; the thick and the dashed lines represent the mean and the mean ± SD, respectively. (**G**) T-Bra polarization (n = 11 aggregates) with respect to the aggregate deformation between 72 and 96 hr; the thick and the dashed lines represent the mean and the mean ± SD, respectively.

The online version of this article includes the following figure supplement(s) for figure 1:

**Figure supplement 1.** Ecad-mGFP/Oct4-mCherry ESC line validation.

---

aggregate; It is present at every cell-cell contact forming a quite homogeneous network of junctions. However, this network undergoes a gradual fragmentation (*Figure 3A* and *Videos 8–11*). The level of E-cad decreases temporally while a small fraction is retained, which eventually localizes near the aggregate's tip. Within our immunostainings, we found consistently the presence of some E-cad/T-Bra cells that express lower levels of E-cad compared to their neighbors devoid of T-Bra (*Figure 3—figure supplement 1A*, 92 hrs-pp). In view of earlier studies (*Turner et al., 2014*; *Fernando et al., 2010*), a gradual repression of E-cad via increasing levels of T-Bra may possibly account for the fall of the E-cad level in the junctional network. Furthermore, in immunostainings at 3 days-pp, we notice only a small fraction of cells – distributed in a salt and pepper manner – that express Foxa2 (*Figure 3—figure supplement 1B*). Therefore, at 72 hr-pp, we do not find an organized group of cells that can potentially yield the endoderm. Additionally, at this stage, the cell population in the aggregates can be broadly classified into three categories: a class of cells that express both E-cad and T-Bra and two classes that express either one of the two markers (*Figure 3B*, left). Additionally, the E-cad+/T-Bra- cells are also found to be the ones expressing Sox2, akin to an epiblast. The E-cad/T-Bra expressing cells constitute most of the cell population (~70% of the total) and are surrounded by the T-Bra expressing cells (*Figure 3B*, right). The cells expressing only E-cad – present in the minority – are present mainly within the core of the E-cad/T-Bra expressing cells (*Figure 3B*, right, inset). A spatial connectivity graph shows that the cells in the core (E-cad+/T-Bra+) are found to be tightly packed – with 70 out of 81 of E-cad+/T-Bra+ cells forming the largest connected component - and maintain a homotypic neighborhood (72 E-cad+/T-Bra+ cells maintain an E-cad+/T-Bra+ neighborhood) (*Figure 3b*, right). Yet again the cells expressing only T-Bra are sparsely wrapped around the core (13 components for 31 T-Bra+ cells) and therefore, have a relatively

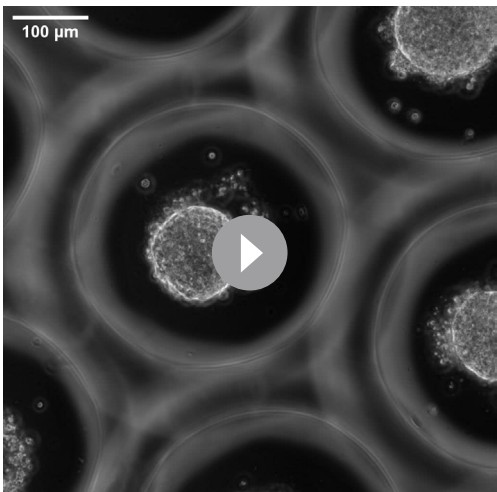

**Video 1.** Gastruloid polarization between 72 hr and 96 hr imaged with bright-field microscopy.
https://elifesciences.org/articles/59371/figures#video1

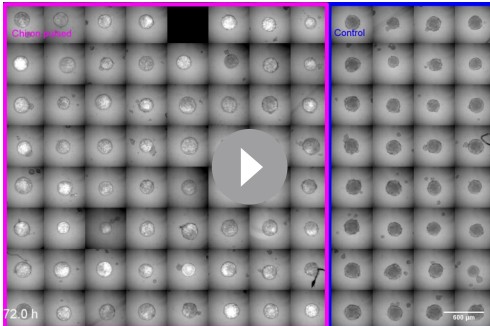

**Video 2.** Gastruloids polarization in a 96 wells plate between 72 hr and 100 hr after formation (mosaic aggregates made of 100 cells E14 +20 cells T-Bra-GFP) imaged by epifluorescence. In magenta, aggregates were pulsed with Chi between 48 hr and 72 hr and have a T-Bra expression and polarization. In blue, control Gastruloids that were not pulsed and do not exhibit T-Bra expression.
https://elifesciences.org/articles/59371/figures#video2

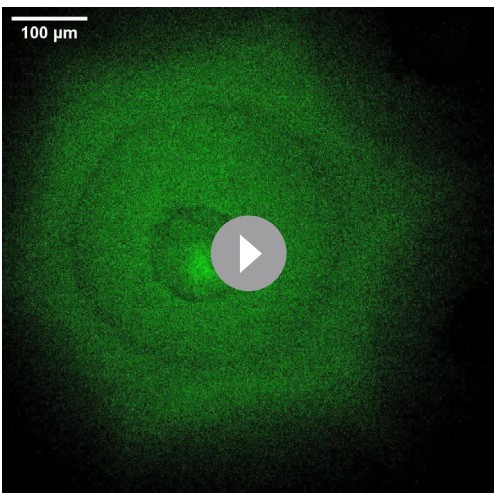

**Video 3.** E-cadherin-GFP signal evolution during Gastruloid polarization between 72 hr and 96 hr imaged by epifluorescence.
https://elifesciences.org/articles/59371/figures#video3

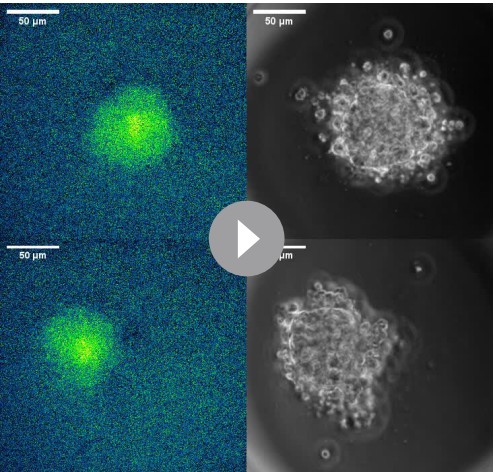

**Video 5.** E-cadherin-GFP progressive decay during the Chi pulse from 48 hr to 72 hr imaged by epifluorescence.
https://elifesciences.org/articles/59371/figures#video5

heterotypic neighborhood (12 T-Bra+ cells have an E-cad+/T-Bra+ majority neighborhood). Spatial distribution of the cell types (mean of n = 7 aggregates) at 72 hrs-pp depicts that the cells expressing only E-cad are mostly present within the E-cad+/T-Bra+ cell islands, whereas the T-Bra+ cells are situated at the periphery of the aggregate (*Figure 3C*). Interestingly, at later time-points but before the aggregates are fully polarized, we observed an 'archipelago' of E-cad expressing cells separated by T-Bra expressing cells (*Figure 3D*, 92 hrs-pp, *Figure 3—figure supplement 1A*). At day 4, the E-cad expressing cells exhibit low levels of N-cadherin and Snail1, supporting the absence of an EMT transition for the E-cad expressing cells (*Figure 3—figure supplements 1C and 2*). These data are consistent with recent observations in the mouse (*Scheibner et al., 2021*). Additionally, the islands of E-cad expressing cells were found to be positive for the pluripotency markers Oct4 (*Figure 3D*). The Oct4 expression correlates inversely with the T-Bra expression, suggesting pluripotency and an epiblast-like nature of these cells (*Figure 3E*). Altogether our data suggests that the endoderm-like region - identified at the end of day 4 – originates from islands of pluripotent and epithelial cells expressing E-cadherin.

## Tissue flows and a local sorting segregate islands of E-cadherin expressing cells

Our observations raise the possibility that the islands of E-cad+ cells sort and form a compact

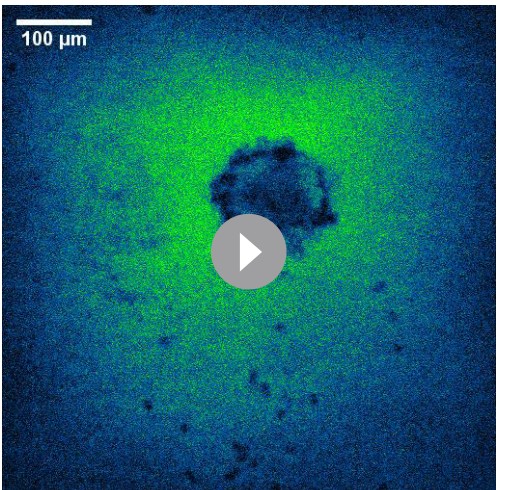

**Video 4.** T-Bra-GFP signal apparition during the Chi pulse from 48 hr to 72 hr imaged by epifluorescence.
https://elifesciences.org/articles/59371/figures#video4

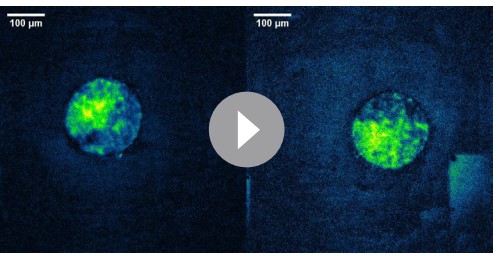

**Video 6.** T-Bra-GFP signal evolution during Gastruloid polarization between 72 hr and 96 hr imaged by epifluorescence.
https://elifesciences.org/articles/59371/figures#video6

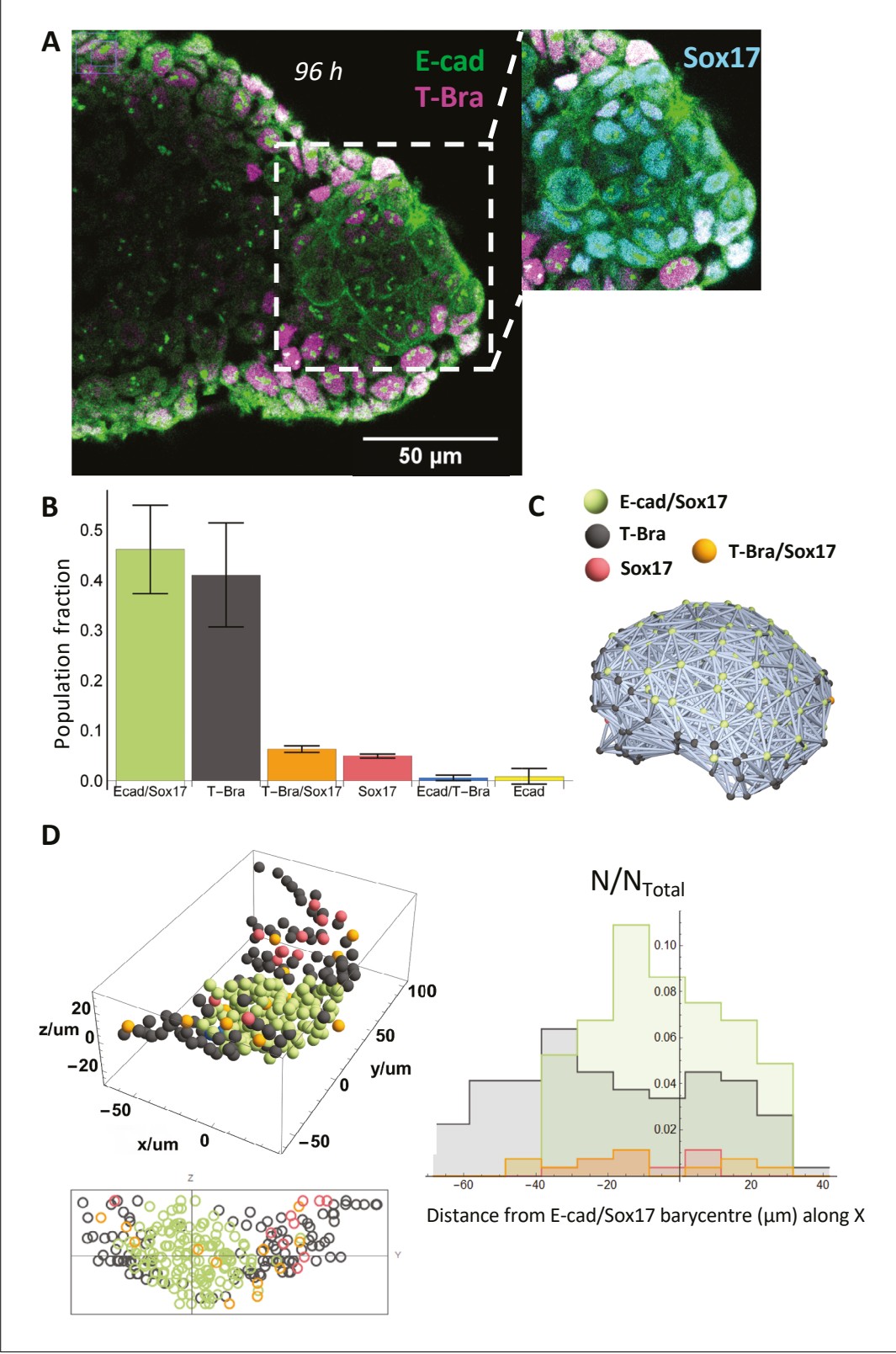

**Figure 2.** An endoderm-like tissue at the tip of the gastruloid. (**A**) aggregate stained at 96 hrs-pp for E-cad and T-Bra; the inset shows the endoderm-like cells at the tip expressing both E-cad and Sox17, surrounded by T-Bra+ cells. (**B**) the bar chart delineates the population fraction of the different cell populations present near the tip (n = 3 aggregates: 268, 179, and 150 cells, N = 2 replicates) with the error bars indicating the SD. (**C**) Spatial connectivity

*Figure 2 continued on next page*

*Figure 2 continued*

graph (Delaunay Mesh) of the different cell populations in an aggregate fixed at 96 hr-pp (**D**) Spatial distribution of the different cell types near the tip of a representative aggregate (stained at 96 hr-pp), its projection and the respective population fractions as a function of the distance from the tip of the aggregate.

The online version of this article includes the following figure supplement(s) for figure 2:

**Figure supplement 1.** Characterization of the endoderm-like region in aggregates at 96 and 120 hrs-pp.

**Figure supplement 2.** Spatial distribution of the different cell types near the tip.

group of cells. To test this possibility, we analyzed cell movements prior to the establishment of the aggregate tip. We imaged the aggregates with inverted two-photon microscopy adding Sulforhodamine B (SRB) to the medium which enables to visualize cell contours and extracellular space. We perform optical flow (See Materials and method) on the SRB signal (*Figure 4A–B*, *Videos 8–14*) and reconstruct the coarse-grained velocity field within the tissue. The flow field clearly shows that there is a directed flow of cells towards the future region that will form the tip of the gastruloid (*Figure 4A–B*, *Figure 4—figure supplement 1*, and *Videos 15–17*). E-cad- cells flow backward at the periphery, away from the tip, which creates a physical separation between the two populations: E-cad+ and E-cad- cells (*Videos 8–10*). Islands of E-cad+ cells are highly dynamic during their movement toward the tip (*Video 11*). The directed flow and recirculation are not observed in absence of Chi pulse (*Figure 4—figure supplement 1*).

Despite dynamic cell rearrangements and shear in the bulk of the aggregate, the islands of E-cad expressing cells are not fully fragmented, implying that physical forces maintain their cohesion (*Video 11*). The E-cad expressing cells in islands have a significantly smaller contact angle with the surrounding cells than with each other (*Figure 4c*, inset). Assuming that the cell-cell contacts are determined by interfacial tension (Supplementary material), we measured the angle of contacts between E-cad expressing cells within the islands and with neighboring cells (*Figure 4C*). They correspond to a significant higher tension at the outer boundary of islands at days 3 and 4 (*Figure 4C*, *Figure 4—figure supplement 2*). Such difference could account for the maintenance of E-cad+ cell islands during their transport as well as their segregation at the tip, where E-cad+ cells would sort from the T-Bra+ cells to form a large endoderm domain.

## Discussion

In contrast to classical gastruloids, when aggregates of ESCs are grown in Activin and FGF, they go through a clear phase similar to the postimplantation epiblast. In this state, suppression of Wnt signaling leads to anterior neural development (*Girgin et al., 2021*) but, as we show here, exposure to Wnt signaling leads to fates associated with primitive streak, as evidenced by expression of T-Bra and the emergence of an endoderm-like region. We also observe directional and coordinated movements of cell ensembles that mimic some aspects of mammalian gastrulation, in particular the epithelial organization of the gastruloid at the time of Chi application. Under Chi exposure, that mimics the Wnt/β-catenin signaling, the pluripotent epiblast-like cells (with the exception of a few cells) experience a massive wave of T-Bra expression and a loss of E-cad-mediated contacts (*Figure 4D*). The cells near the periphery specifically exhibit high levels of T-Bra and low levels of E-cad expression (*Figure 3B and C*). A similar observation has been made on 2D micropatterns in which a T-Bra wave emanates and travels inwards, from the boundary where E-cad is absent (*Martyn et al., 2019*). Loss of

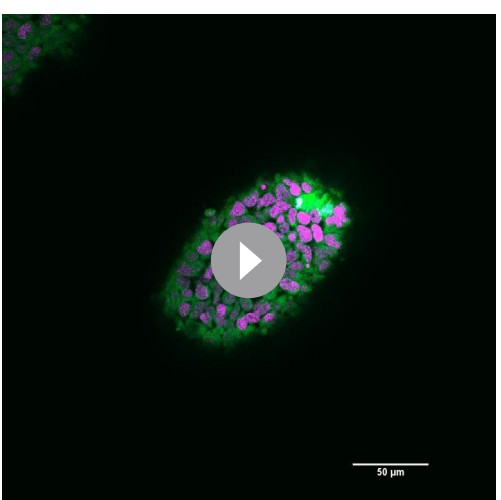

**Video 7.** Z-stack (one slice every 10 microns) of a 96 hr Gastruloid (made of 150 T-Bra-GFP cells) immunostained with FoxA2 and E-cadherin.
https://elifesciences.org/articles/59371/figures#video7

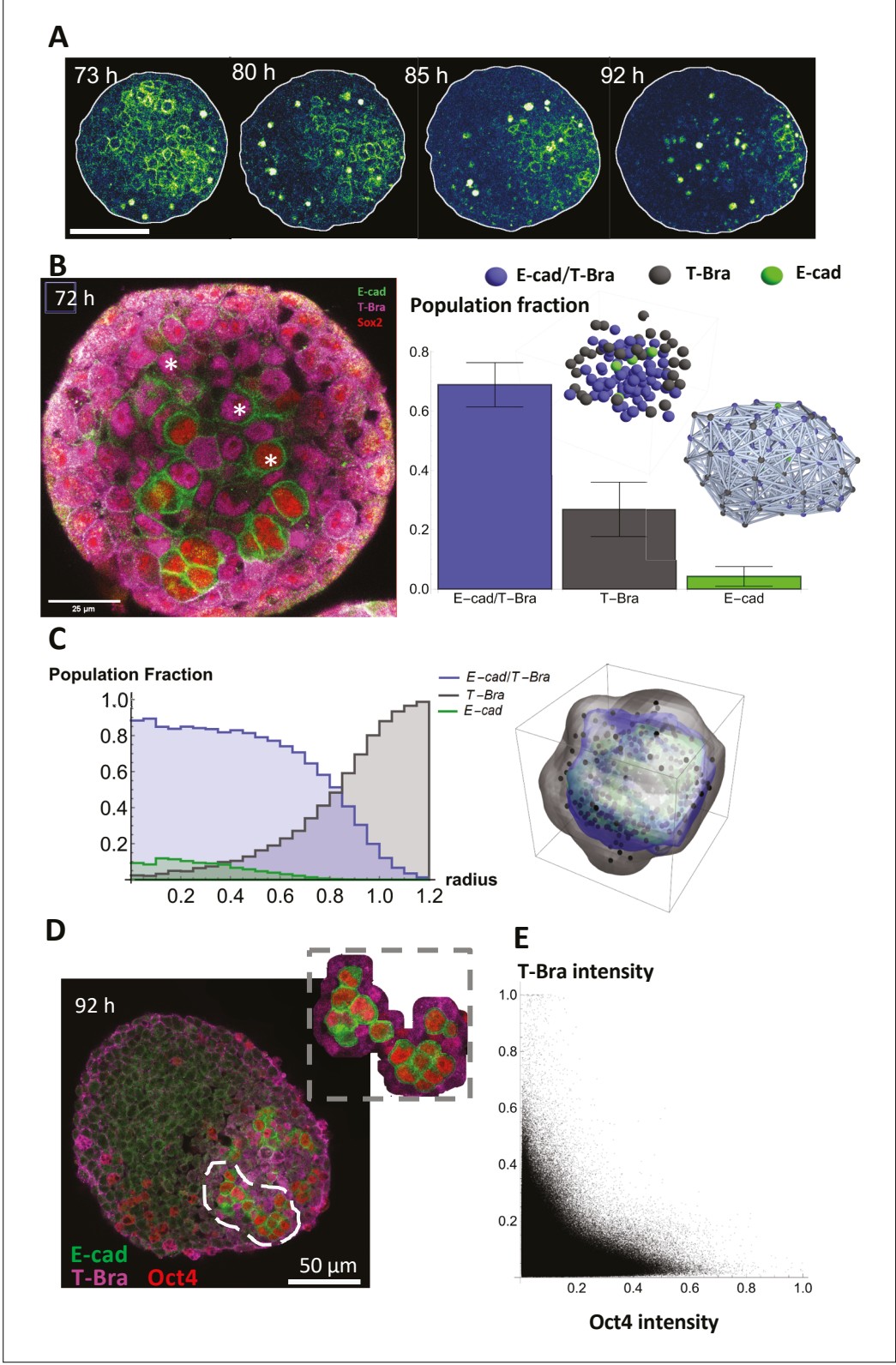

**Figure 3.** The endoderm-like region originates from islands of pluripotent and epithelial cells expressing E-cadherin. (**A**) Time lapse images showing a gradual fragmentation and loss of E-cad network at 73, 80, 85, and 92 hrs-pp; scale bar represents 100 μm (**B**) (left) The three different cell populations in the aggregate are marked at 72 hrs-pp: namely cells that express only T-Bra, cells that express both T-Bra and E-cad, and cells that express only

*Figure 3 continued on next page*

*Figure 3 continued*

E-cad. (right) the fraction of the total population of cells for the three classes are provided in the histogram (n = 7 aggregates: 126, 116, 121, 102, 85, 184, and 163 cells, N = 2 replicates) with the SD shown by the error bars; inset shows the spatial distribution of the cells in one such aggregate (right) Spatial connectivity graph is generated for the aggregate at 72 hrs-pp. (**C**) Spatial distribution of the population fraction of the different cell types as a function of the radial distance from the aggregate centre (n = 7 aggregates: 126, 116, 121, 102, 85, 184, and 163 cells, N = 2 replicates). (**D**) Islands of E-cad/Oct4 expressing cells surrounded by T-Bra+ cells are present within the aggregate fixed at 92 hr-pp; the inset zooms on a few such islands. (**E**) Pixel intensity correlation plot between Oct4 and T-Bra (n = 6 aggregates, N = 2 replicates).

The online version of this article includes the following figure supplement(s) for figure 3:

**Figure supplement 1.** Immunostaining for E-cadherin, N-cadherin, T-Bra, Sox2, Foxa2 in aggregates at different stages.

**Figure supplement 2.** Immunostaining of E-cadherin and Snail in gastruloids at 66 hr, 72 hr, and 90 hr (n = 5 aggregates, N = 1 replicate).

E-cad contacts in parts of the aggregate leads to the appearance of pluripotent E-cad+ cell islands (Sox2 or Oct4 positive), that are enveloped by T-Bra expressing cells (*Figure 3*, B, D, *Figure 3—figure supplement 1*, and *Figure 4D*). All the while a coordinated motion of the cells within the aggregate arrange the islands of the E-cad/Sox2/Oct4 expressing cells at one pole of the aggregate. This pole protrudes outwards to form the tip. The E-cad islands subsequently segregate at the tip and are found to express markers characteristic of the endoderm (*Figures 2 and 4D*).

While we cannot rule out the possibility that the mechanogenetic trajectory followed by cells in the gastruloids may be different from that in the embryo, our data are consistent with recent observations in the embryo (*Scheibner et al., 2021*) that challenged the prevailing view that endoderm forms from a mesendodermal primodium. In the mouse (*Scheibner et al., 2021*) and in the gastruloids (our work and *Vianello and Lutolf, 2020*) the endodermal progenitors do not undergo an Epithelial-Mesenchymal transition (EMT) and emerge from a different pool of cells (E-cad+ /T-Bra-) than the mesodermal-like precursor cells (T-Bra+). Vianello and Lutolf further reported that within 7 days the endoderm primordium in the gastruloids give rise to different expression territories corresponding to anterior foregut, midgut and hindgut (*Vianello and Lutolf, 2020*), suggesting that what we observe in vitro follows closely the events in vivo. Altogether the in vivo and in vitro data point to a mechanism where the endoderm tissues originate from the epiblast via a trajectory that differs from that of the mesodermal cells.

A surprising feature of endoderm specification in gastruloids (see also *Pour et al., 2019*; *Vianello and Lutolf, 2020*) is the appearance of endodermal cells as small islands that coalesce into a continuous tissue. The evidence that a similar mechanism is at play in the mouse embryo remains partial, although a recent report suggests that endodermal progenitors emerge in small islands (see Figure 3 in *Probst et al., 2021*). In zebrafish, a similar mode of endoderm specification has been reported, where it is the predominant mode (*Montero et al., 2005*; *Nair and Schilling, 2008*; *Warga and Nüsslein-Volhard, 1999*). The clear manifestation of this mode of specification in the gastruloids

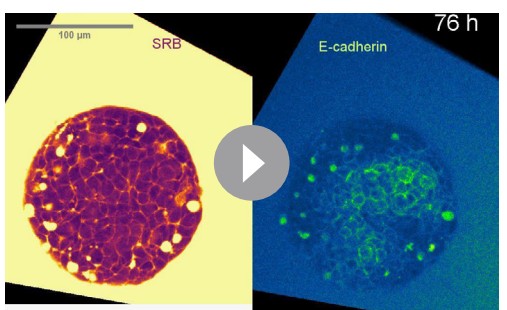

**Video 8.** Two-photon time-lapse movie of Gastruloid polarization and elongation with E-cadherin-GFP and sulforhodamine B (SRB) tagging.
https://elifesciences.org/articles/59371/figures#video8

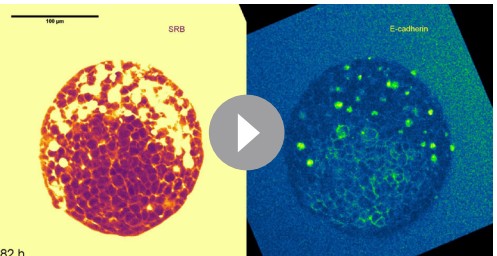

**Video 9.** Two-photon time-lapse movie of Gastruloid polarization and elongation with E-cadherin-GFP and sulforhodamine B (SRB) tagging.
https://elifesciences.org/articles/59371/figures#video9

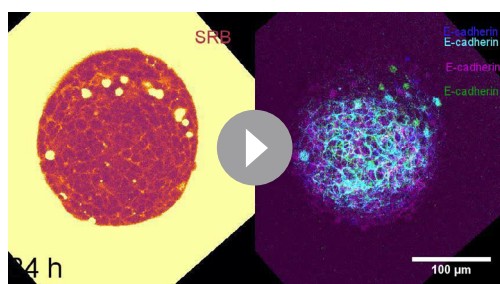

**Video 10.** Two-photon time-lapse movie of Gastruloid polarization and elongation with E-cadherin-GFP and sulforhodamine B (SRB) tagging. Four planes 12.5 µm apart are superimposed on the right panel.
https://elifesciences.org/articles/59371/figures#video10

might reflect that ancestral modes are emphasized in this model (*Steventon et al., 2021*).

The segregation of endodermal-like cells in the gastruloids is a consequence of a wave of cell state transitions, a global flow and surface tension heterogeneity. Whether the same mechanism exists at the primitive streak in the mouse embryo remains to be explored. The mechanism described here cannot be reduced to a simple cell-sorting mechanism based on differential adhesion. Different cellular mechanisms are at stake: advection by tissue flow, cell sorting by cellular motions, but also cell division, cell death, and cell differentiation that constantly modify the number of constituent cells and their pattern. How these different cell events lead to the eventual tissue pattern will require further study. Our work emphasizes the coupling between biochemical signaling and tissue flows during endoderm formation, and perhaps more generally during the morphogenesis of germ layers. Furthermore, it illustrates the potential of in vitro models of early embryogenesis to decouple and study these events.

## Materials and methods
### Cell culture and gastruloids
A complete description of the culture conditions and the protocol for making gastruloids is present in *Baillie-Johnson et al., 2015*. Nevertheless, a twofold modification was made to the original protocol.

a. Unless explicitly mentioned approximately 50–60 cells were seeded per well to generate smaller gastruloids.
b. The gastruloids were exposed to a 12.5 ng/ml of bFGF (RD system ref: SUN11602) and 25 ng/ml of Activin A (RD system ref: 338-AC-050) throughout its development

 * We formed our aggregates in low-adherence condition in 96 well plates (Costar ref: 7007) and microwell plates (400 um wells, SUNBIOSCIENCE ref: Gri3D).

 List of cell lines used:

1. the new E-cad-GFP/Oct4-mCherry cell line is described in *Figure 1—figure supplement 1*.
2. T-Bra-GFP/NE-mKate2: the TBra-GFP/NE-mKate2 was generated in Sally Lowell group through a random integration of a NE-mKate2 plasmid (*Blin et al., 2019*) in a TBra-GFP cell line developed in Val Wilson group (*Karagianni, 2017*).The cell line was authenticated by Southern blot.
3. E14Tg2a.4 mouse Embryonic Stem Cells (mESC) (MMRRC, University of California Davis, US).

All cell lines were tested free from mycoplasma contamination by qPCR.

### Immunostaining

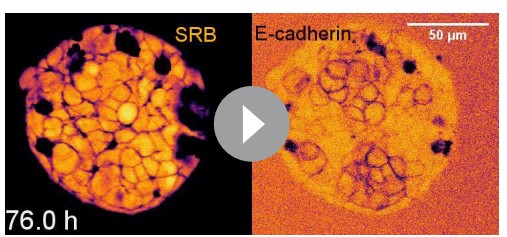

**Video 11.** Two-photon rapid time-lapse movie of a Gastruloid at early phase of symmetry breaking (E-cadherin-GFP).
https://elifesciences.org/articles/59371/figures#video11

Immunostaining was performed by fixing the aggregates (4% PFA in PBS with 1 mM $CaCl_2$ and 0.49 mM $MgCl_2$ overnight at 4 °C), followed by washing and blocking for 1 hour in PBSFT buffer (PBS with 10% FBS, 0.2% Triton X-100). Aggregates were incubated with primary antibodies, washed, and incubated again with secondary antibodies. The incubations were done overnight at 4 °C.

Mentioned below is a list of primary antibodies used in this study:

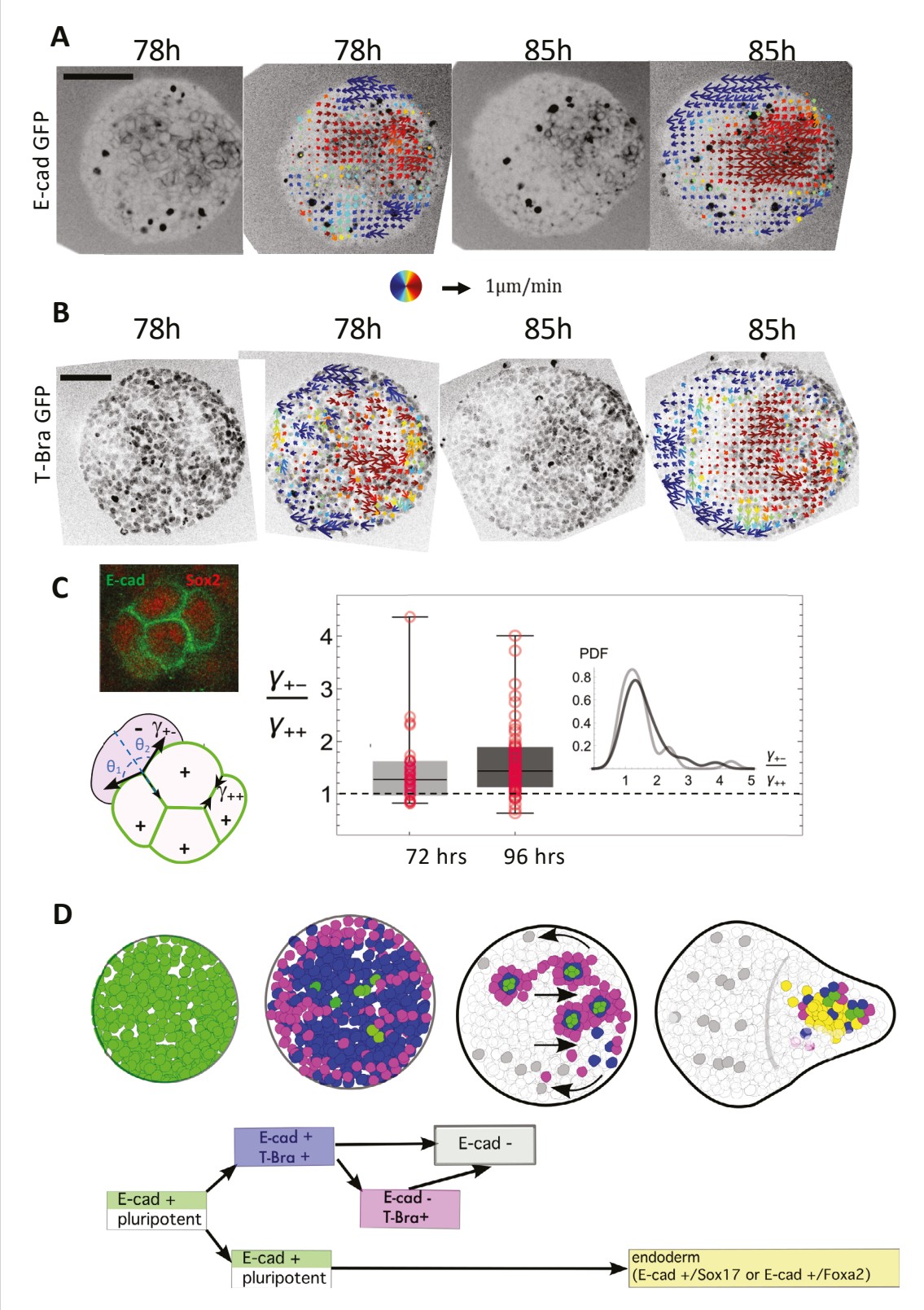

**Figure 4.** Tissue flows and a local sorting segregate islands of E-cadherin expressing cells. (**A–B**) Cellular movements are quantified during the aggregate elongation using two-photon imaging of the aggregates with Sulforhodamine B (SRB) added in the medium, which enables to visualize cell contours. Optic flow measurements are performed on the SRB signal and the obtained velocity fields overlapped on the GFP signal corresponding to (**A**) E-cad and (**B**) T-Bra (N = 3 replicates). (**C**) Box-whisker plot shows the distribution of the ratio of the cell-junction tension heterogeneity between

*Figure 4 continued on next page*

*Figure 4 continued*

E-cad expressing cells (+), and between E-cad expressing cells and their neighbors (-) at 72 hr-pp (n = 28 junctions, N = 2 replicates) and 96 hr-pp (n = 54 junctions, N = 2 replicates); 25th and 75th quartiles together with the minimum and maximum values are shown; the insets to the left depict regions on the E-cad+ cell islands where the angle measurements were made. No significant statistical difference between the two time points (Kolmogorov-Smirnov Test; p-value = 0.37) (**D**) Cartoon depicting the different steps leading to the formation of an endoderm-like region within the gastruloid. Upon addition of Chi, majority of the E-cad expressing cells begin to express T-Bra. The transient expression of T-Bra represses E-cad expression in T-Bra+/E-cad+. Formation of islands of E-cad expressing cells surrounded by T-Bra+ cells; these islands are transported to one end of the aggregate via a tissue-scale movement. Islands of E-cad expressing cells segregate to give rise to the endoderm.

The online version of this article includes the following figure supplement(s) for figure 4:

**Figure supplement 1.** Characterization of tissue flows.

**Figure supplement 2.** Islands of E-cadherin expressing cells.

| Antibodies | Species | Reference | Provider | Dilution |
|---|---|---|---|---|
| Brachyury | Goat | AF2085 | RD system | 1:80 |
| E-cadherin | Rat | M108 | Takara | 1:200 |
| anti-GFP | Chicken | GFP-1020 | Aves | 1:1000 |
| Sox17 | Goat | AF1924 | RD system | 1:80 |
| FoxA2 | Goat | sc-6554 | Santa-Cruz | 1:50 |
| CDX2 | Rabbit | MA514494 | Thermo | 1:200 |
| Gata6 | Goat | AF1700 | RD system | 1:100 |
| Sox2 | Mouse | Sc-365823 | Santa Cruz | 1:100 |
| Snail | Goat | AF3639 | RD system | 1:50 |

*We used Alexa dyes (ThermoFisher) against the primary antibodies.

## Live and fixed imaging

Epifluorescence time-lapse imaging of gastruloids was achieved in a chamber – maintained at 37 °C, 5% $CO_2$ with a humidifier – using an AxioObserver inverted microscope (Carl Zeiss) with a 20 x LD Plan-Neofluar 0.4 NA Ph2 objective. A white LED source (Laser2000, Kettering UK) was used for illumination. A second setup with similar specifications was also utilized for long-term imaging.

Fixed gastruloids were loaded on MatTek dishes (MatTek corporation, ref: P35G-1.5–14 C) and imaged using a confocal microscope (Zeiss LSM 880) with a 40 × 1.2 NA W Korr objective.

Biphoton imaging of gastruloids was achieved in a chamber- maintained at 37 °C, 5% CO2 with a humidifier – using a Zeiss 510 NLO (Inverse - LSM) with a femtosecond laser (Laser Mai Tai DeepSee HP) (900 nm) with a 40 x/1.2 C Apochromat objective. Seventy-two hr old gastruloids were transferred from 96-well plates to either MatTek dishes (MatTek corporation, ref: P35G-1.5–14 C) or to microwell plates (400 um wells, SUNBIOSCIENCE ref: Gri3D). To visualize intercellular space, we added 32 µL/mL of a Sulforhodamin B solution (1 mg of Sulforhodamin B powder, 230,162 Aldrich per mL of medium) to the medium. We imaged sub-volumes

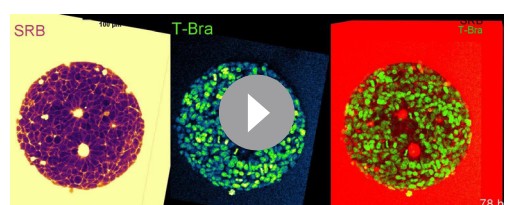

**Video 12.** Two-photon time-lapse movie of Gastruloid polarization and elongation with T-Bra-GFP and sulforhodamine B (SRB) tagging.

https://elifesciences.org/articles/59371/figures#video12

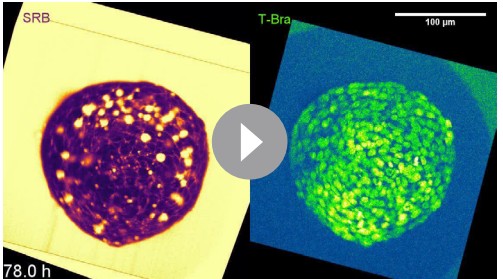

**Video 13.** Two-photon time-lapse movie of Gastruloid polarization and elongation with E-cadherin GFP and sulforhodamine B (SRB) tagging.

https://elifesciences.org/articles/59371/figures#video13

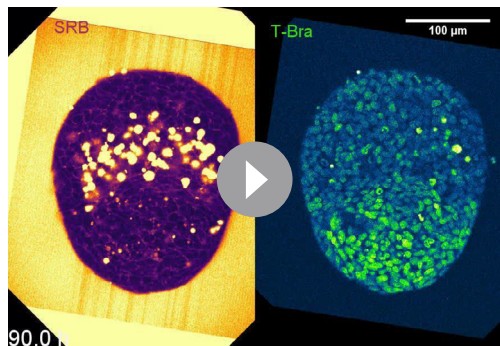

**Video 14.** Two-photon time-lapse movie of Gastruloid polarization and elongation with T-Bra::GFP and sulforhodamine B (SRB) tagging.

https://elifesciences.org/articles/59371/figures#video14

of gastruloids (10 slices z-stacks with a z-interval of 2 µm) near their midplane during a total time between 10 h and 24 h (time interval between two frames between 2 and 4 min depending on the experiment). We imaged the GFP (T-Bra-GFP/ NE-mKate2 and E-cadherin-GFP/Oct4-mCherry cell lines) and the Sulforhodamin B channels using a non-descanned detector.

## Cryosection

Aggregates were embedded in Optimal Cutting Temperature (OCT) after fixing in 4% Paraformaldehyde (PFA) and then flash frozen in liquid nitrogen. Slices of 25 and 30 µm were manually cut from the resin with a cryostat (Leica CM3050 S). The slices were mounted on Superfrost Plus slides before proceeding with immunostaining.

## Relative surface tension

Under the assumption that the final shape and configuration of the cells are largely dominated by surface tension, we can compute the relative

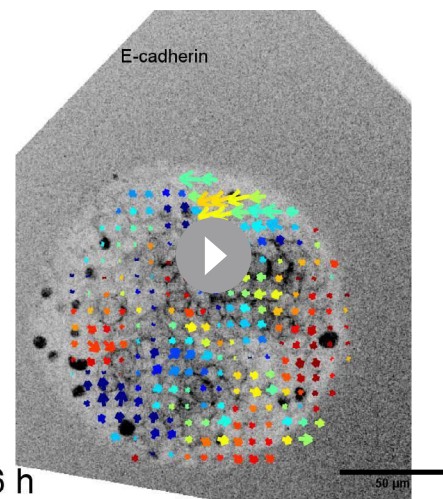

**Video 16.** Velocity field obtained by optical flow on the sulforhodamine B signal of the movie 8 superimposed on the E-cadherin signal.

https://elifesciences.org/articles/59371/figures#video16

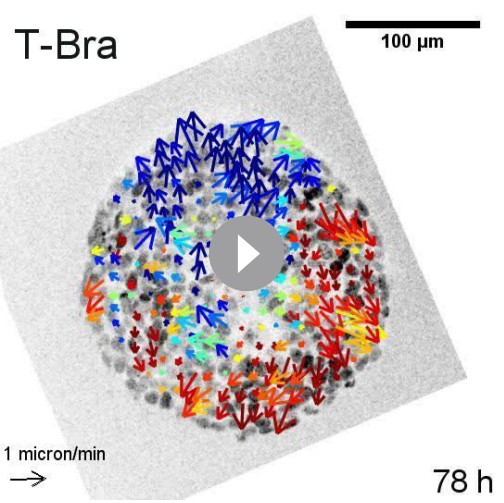

**Video 15.** Velocity field obtained by optical flow on the sulforhodamine B signal of the movie 12 superimposed on the T-Bra signal.

https://elifesciences.org/articles/59371/figures#video15

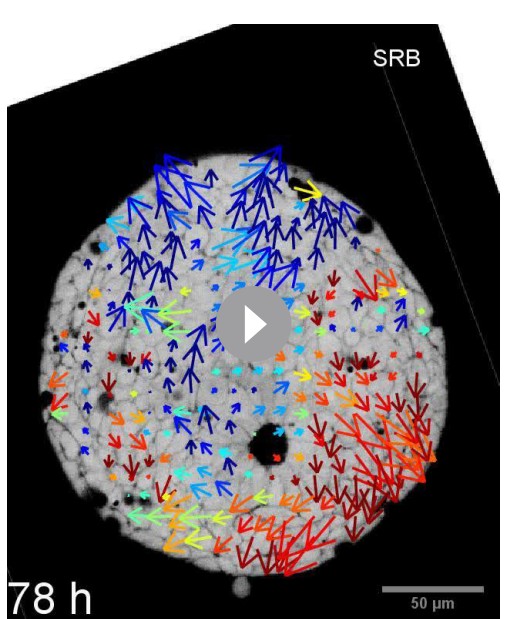

**Video 17.** Velocity field obtained by optical flow on the sulforhodamine B signal of the movie 12 superimposed on the sulforhodamine B signal.

https://elifesciences.org/articles/59371/figures#video17

tension between the E-cadherin/non-E-cadherin $\left(\gamma_{+-}\right)$ and the E-cadherin/E-cadherin expressing cells $\left(\gamma_{++}\right)$ as the reciprocal of the sum of the cosines of the two angles subtended at the interface of the E-cadherin/non-E-cadherin cells (*Figure 4C*).

$$\left(\frac{\gamma_{+-}}{\gamma_{++}}\right) = \frac{1}{cos\ \theta_1\ + cos\ \theta_2}$$

## Image processing and data analysis triangulation/probability map

The gastruloids were initially manually segmented using brightfield images or a combination of bright-field and the fluorescence signal. An artificial convolutional neural network, UNET (*Ronneberger et al., 2015*) was also employed to yield a high-throughput pipeline for generating binary masks of aggregates with brightfield images. The UNET was trained on an augmented dataset (n = 390 and n = 264 images for aggregates generated from Ecad/Oct4 line and T-Bra/NE-mKate2 lines respectively) and then applied onto the time series aggregate images to obtain binarized masks. Any imperfections in the generated masks were manually corrected in FIJI/ImageJ. The mean signal intensity of the aggregates was computed by averaging the pixel intensities within the binary masks.

The overall elongation of the aggregates at each time point was quantified by determining the ratio of the eigenvalue pairs of the inertia tensor of the binary mask. This is equivalent to finding the ratio of the major to the minor axes of the best-fit ellipse to the mask. Furthermore, to precisely capture the onset of the tip formation we relied on Fourier-Elliptic Descriptors[3](FED) as a shape descriptor (*Kuhl and Giardina, 1982*). The aggregate perimeter was defined by a set of n-harmonics/ellipses (n = 1,2,3 …) – each associated with four coefficients – such that the difference of area between the aggregate mask and the morphology obtained from the harmonics was less than 2%. The onset of tip formation was determined using the 2nd FED harmonic at each time point. The corresponding coefficients were transformed to the lengths of the major and minor axes of the ellipse. The magnitude of the second harmonic was then taken to be the product of the lengths of the two axes, which is a proxy for the extent of tip formation and shape polarization.

The polarization of the fluorescence was defined as the maximum of contrast between the two-halves of the aggregate. It was calculated in two-steps: first the aggregate mask was chopped into two halves about the eigenvector of the inertia matrix, which passes through the centroid of the mask. The contrast was then calculated as (I_max-I_min)/(I_max + I_min) where I_max and I_min represent the intensities of the brighter and less bright halves, respectively. The contrast maximum was then determined by angular scanning by repeatedly rotating the cutting plane by 5°.

The different cell-types in the confocal image-stacks were manually identified and tagged via *Cell Counter* – a built-in FIJI plugin. Corrections were made to ensure that each cell was designated a unique identity. The positions of the marked cells were extracted for further processing. The centroids of the cell were extracted to create a neighborhood graph and deduce spatial statistics. The cell populations at the aggregate tip (at 96 hr-pp) are spatially binned to generate probability maps of the different cell types along the tip. Owing to the sphericity of the aggregates at 72 hrs-pp, cell positions are normalized to lie in a unit sphere. Aggregates are registered via their common origin and all the cell populations are pooled to create a spatial probability density map (using SmoothKernelDistribution function in Mathematica). A numerical integration of the density function yields the probability of finding the various cell types as a function of the distance from the origin.

To build the graph we generated a Delaunay triangulation (using the built-in DelaunayMesh in Mathematica) from the cell centroids and pruned any edges that were more than 12 um in length (avg. size of a cell). The neighborhood type of a cell is assigned by counting the cell types of all the adjoining cells and taking the most dominant cell-type as the neighborhood type; for instance, if majority of the neighboring cells, around an E-cad+/T-Bra+ cell, are also E-cad+/T-Bra+ then the cell has a homotypic neighborhood, else the neighborhood is designated as being heterotypic.

To measure velocity fields corresponding to Gastruloids polarization and elongation, we use the signal obtained from the Sulforhodamin B (SRB) channel in two-photon timelapses (which is present in all the parts of the Gastruloid in contrary to signals such as T-Bra GFP or E-Cadherin GFP). The SRB signal corresponds to (i) cell contours, (2) dead cells, and (3) bright spots of subcellular size moving rapidly corresponding probably to vesicles are visible. As dead cells and bright spots are much brighter than cell contours, we filter them with a thresholding operation, and perform the optic flow on the filtered image where only cell contours are visible after timelapse registration to remove rapid

rotations and translations of the aggregate. We use a custom-made Matlab optic flow code based on the Kanade Lucas Tomasi (KLT) algorithm with a level 2 pyramid and a window size corresponding to a square of 2 cells by 2 cells (*Lucas and Kanade, 1981*).

Images in the timeseries data were registered using *MultiStackReg* – an ImageJ plugin – and custom Matlab scripts. The scripts for analyzing data, image processing, neural networks and computer vision techniques were implemented in Wolfram Mathematica 12 and Matlab. Analysis scripts are provided in the supplemental information of this paper.

## Acknowledgements

The work is supported by the Leverhulme Trust (visiting professorship to PFL VP2-2015-022 and RPG-2018–356), Agence Nationale de la Recherche ("Investissements d'Avenir", Labex INFORM Phd Grant to AH, ANR-11-LABX-0054 and ANR-16-CONV-0001 from Excellence Initiative of Aix-Marseille University - A*MIDEX and generic grant to PFL, ANR-19-CE13-0022), the Lundbeckfonden (Lundbeck Foundation, R198-2015-412). The Novo Nordisk Foundation Center for Stem Cell Medicine is supported by a Novo Nordisk Foundation grant (NNF21CC0073729). The Novo Nordisk Foundation Center for Stem Cell Biology was supported by a Novo Nordisk Foundation grant (NNF17CC0027852). We also acknowledge the France-Bioimaging Infrastructure (ANR-10-INBS-04). We thank Simone Probst and Sebastian Arnold for their advice. We are grateful to Rosanna Dono for her suggestions and help, and Tina Balayo for assistance with tissue culture. We thank all members of the Lenne and Martinez Arias groups. We thank Guillaume Blin, Sally Lowell and Val Wilson (CRG, University of Edinburgh) for sharing their cell line.

## Additional information

### Funding

| Funder | Grant reference number | Author |
|---|---|---|
| Agence Nationale de la Recherche | ANR-19-CE13-0022 | Pierre-François Lenne |
| Agence Nationale de la Recherche | ANR-11-LABX-0054 | Pierre-François Lenne<br>Ali Hashmi |
| Agence Nationale de la Recherche | ANR-16-CONV-0001 | Pierre-François Lenne<br>Sham Tlili |
| Agence Nationale de la Recherche | ANR-10-INBS-04 | Pierre-François Lenne<br>Sham Tlili<br>Ali Hashmi |
| Leverhulme Trust | VP2-2015-022 | Pierre-François Lenne<br>Alfonso Martínez Arias |
| Leverhulme Trust | RPG- 2018-356 | Pierre-François Lenne<br>Alfonso Martínez Arias |
| Lundbeckfonden | R198-2015-412 | Joshua M Brickman |
| European Research Council | AdG MiniEmbryoBlueprint 834580 | Alfonso Martínez Arias |

The funders had no role in study design, data collection and interpretation, or the decision to submit the work for publication.

### Author contributions

Ali Hashmi, Conceptualization, Formal analysis, Investigation, Methodology, Software, Writing - original draft, Writing - review and editing; Sham Tlili, Investigation, Methodology, Software, Writing - review and editing; Pierre Perrin, Investigation; Molly Lowndes, Hanna Peradziryi, Investigation, Resources; Joshua M Brickman, Funding acquisition, Resources, Supervision; Alfonso Martínez Arias, Conceptualization, Funding acquisition, Writing - review and editing; Pierre-François Lenne, Conceptualization, Formal analysis, Funding acquisition, Investigation, Methodology, Supervision, Writing - original draft, Writing - review and editing

## Author ORCIDs

Ali Hashmi http://orcid.org/0000-0001-9946-7679
Sham Tlili http://orcid.org/0000-0001-6018-9923
Joshua M Brickman http://orcid.org/0000-0003-1580-7491
Alfonso Martínez Arias http://orcid.org/0000-0002-1781-564X
Pierre-François Lenne http://orcid.org/0000-0003-1066-7506

## Decision letter and Author response

Decision letter https://doi.org/10.7554/eLife.59371.sa1
Author response https://doi.org/10.7554/eLife.59371.sa2

## Additional files

### Supplementary files

• Supplementary file 1. Table – Statistics: number of aggregates analyzed, number of independent experiments and statistical tests.

• Transparent reporting form

### Data availability

All data generated or analysed during this study are included in the manuscript and supporting files. The source data and the scripts at https://zenodo.org/record/5727050.

The following dataset was generated:

| Author(s) | Year | Dataset title | Dataset URL | Database and Identifier |
|---|---|---|---|---|
| Hashmi A, Tlili S, Perrin P, Martinez-Arias A | 2020 | Cell-state transitions and collective cell movement generate an endoderm-like region in gastruloids | https://zenodo.org/record/5727050 | Zenodo, 10.5281/zenodo/5727050 |

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
