## [Editor Report]

The authors took advantage of the gastruloid system to explore mechanisms of endoderm specification at cellular resolution. First, they confirm that contrary to mesoderm, nascent endoderm does not undergo epithelial-mesenchymal transition. Second, they provide evidence for a role of tissue flow, and possibly heterogeneity of cellular junction tension, in the sorting and differentiation of the islets of endoderm cells.

---

## [Decision Letter]

**Decision letter after peer review:**

Thank you for sending your article entitled "Cell-state transitions and collective cell movement generate an endoderm-like region in gastruloids" for peer review at *eLife*. Your article is being evaluated by 3 peer reviewers, including Isabelle Migeotte as the Reviewing Editor and Reviewer #1, and the evaluation is being overseen by Didier Stainier as the Senior Editor.

The reviewers agree that the manuscript reports an interesting and original observation in gastruloids. However, there is currently no evidence to propose that such mechanism would be present in embryos. Additionally, there is consensus that the methods are not sufficiently explained, the reproducibility is not clearly quantified, and some claims would require a larger number of aggregates/cells to be solid.

*Reviewer #1:*

General assessment

The manuscript by Hashmi et al., describes the emergence of endoderm-like cells in a stem cells based embryo model. The particularity of the protocol is that it stimulates transition through an epiblast-like state, then differentiation towards mesoderm after a pulse of Chiron, a Wnt agonist. In those conditions, islets of E-cadherin positive cells emerge, surrounded by Ecad+Brachyury+, then Brachyury positive cells. Those islets fuse together at the tip, possibly due to distinct surface tension and directed cell movements, and express endoderm markers such as Sox17 and FoxA2.

It is an original approach and concept, raising new questions and possibilities about the mechanisms of endoderm emergence in the mouse embryo. The manuscript is well written and clear.

Concerns

1.The data would benefit from increased clarity in stating, for each experiment, the proportion of aggregates in which a given phenomenon was observed, as well as the number of cells counted in each aggregate, in particular in supplementary figures.

2. For the migration analysis, it could be interesting to distinguish each cell trajectory in order to distinguish behaviours of the subpopulations.

3. In terms of the surface tension analysis, performing a similar analysis at different timepoints might be helpful to understand how the islets come to fuse at the tip.

4. I am not sure about the specificity of the gata6 staining, not that it adds a lot to the story.

5. The authors might want to discuss how those aggregates evolve, and whether the endoderm-like cells have a potential for further differentiation.

Conclusion

Overall it is an interesting and original observation, well substantiated. More details on the quantification methods would help convince about the solidity of the model: chance of obtaining those cells, amount of cells of each subpopulation including those described in supplementary figures, technical possibility of sorting them for transcriptome analysis etc. In terms of format, I think this report would better fit as a short report than full research article.

*Reviewer #2:*

In this manuscript, the authors proposed a new mechanism of endoderm formation in 3D gastruloid models based on cell migration and fragmentation. Specifically, they found that E-cad is first uniformly expressed inside mESC aggregates. After exposure to Wnt agonist Chiron (Chi), a gradual repression of E-cad and an increase of T-Bra were detected. Cells in the core are tightly packed and express E-cad. T-Bra expressing cells are sparsely wrapped around the core. A directed flow of E-cad expressing cell islands surrounded by T-Bra expressing cells help to accumulate E-cad expressing cells to the tip of the aggregate and form endoderm domain.

I think the dynamical expression of E-cad and T-Bra and the directed cell flow reported in this manuscript are interesting. The results and videos have showed that the elongation and formation of endoderm region is a collective cell behavior rather than single cells undergo epithelial-to-mensenchymal transition. But I am not convinced that the process is done based on the three-step mechanism proposed by the authors. Moreover, I am not sure if this phenomenon really happened in mouse embryo development, giving the considerable differences between gastruloid model and embryo. Since there are methods culturing mouse embryo in vitro up to the early organogenesis stage, I would suggest the authors to provide more evidence showing that the proposed mechanisms might also happen in vivo.

In addition, the manuscript provides too little information to understand the phenomenon. And they did not clearly introduce experimental and computational methods they used to acquire the results. I listed some of my comments below.

1. Did all 3D aggregates become elongated shape in the presence of Chi? If not, what do E-cad and T-Bra expressions and cell migration dynamics look like in those spherical aggregates? Without Chi, inside the spherical aggregates, do they also have cell migration since the aggregates keep growing larger?

2. When did the collective cell migration start? Right after exposing to Chi? Or after some percentage of cells become T-Bra positive cells? Did the gastruloid keep elongating with directed cell flow inside it when cultured for a long time?

3. Are the collective cell migration driven by the T-Bra cells? Is it a spontaneous property of E-cad cells when the E-cad cell density exceed some critical threshold (e.g. glassy dynamics)?

4. Does the elongation and migration dynamics depend on the concentration of Chi, size of the aggregates? I noticed the authors used different initial seeding densities.

5. For the elongated cell aggregates, one side of cells express E-cad. How about the other side of cells? Did they all become mesoderm-like (T-Bra+) cells?

6. Many results are only based on several (3 or 4) gastruloids. For example, figure 1 (d) (e), figure 2 (b), figure 3(c). And in Figure 4 (b), the authors only quantify 13 junctions, probably in the same gastruloid. Due to the heterogeneity among the gastruloids, I am not sure how repeatable the experiments are. Can those observations really reflect phenomenon happened inside the majority of gastruloids? I think the authors should provide some quantifications of the percentage of observing the reported results among a large number of gastruloids.

Unclear results or experimental descriptions:

1. Can the authors show a schematic of the experimental process, such as the time of adding Chi and fixation?

2. 'We find that 30/37 … set to 0.125.' How did the authors define and calculate the elongation ratio and E-cadherin polarization ratio? How did the authors define the elongation threshold?

3. Figure 1 (a): what is the y axis? 1 (d): how did the author measure the E-cad and T-Bra expression? Fixing at different time points or live imaging? If it is live imaging, is the acquisition process influenced by adding and removing Chi? 1(e) how can the authors get continuous results for polarization?

4. Figure 2 (b) Are those dots represents the nuclear position? Can the authors provide the 3D view of the whole gastruloid? (c) What information the authors are trying to get from the connectivity graph?

5. Figure 3 (a) What are those white dots in the images, also in video 6? Can the authors replace t1, t2, t3, t4 with the real time, such as 24h, 36h? (d) How did the authors calculate the intensity? How did the authors normalize the intensity? The schematic in (b) is hard to understand. What do the light and dark colors represent? How did the authors measure theta_1 and theta_2, especially in 3D situation? More quantitative information should be acquired from (a).

6. I am not able to identify islands of E-cad expressing cells in Figure 3 (a) and video 6.

*Reviewer #3:*

In this manuscript Hashmi et al., describe the emergence of an endoderm population in a gastruloid model. They observe that endoderm cells are positive for E-Cad, likely express E-Cad continuously from an epiblast state, initially form small islands, and finally coalesce into a larger endoderm region at the pole of the gastruloid. There are several issues with this manuscript in its current form.

1. No evidence is provided that there is an relationship between how endoderm forms in this gastruloid model and in vivo. In fact, endoderm is believed to derive from a restricted area in the anterior primitive steak. This is evident from the mouse imaging data of Mcdole et al., Cell 2018 as well as from more recent genetic labeling experiments (Probst et al., bioRxiv 2020). It is well known that cells of different germ layers may self-segregate and this may drive the behavior observed here downstream of heterogenous differentiation in the gastruloids, but that is not necessarily the mechanism which occurs in vivo. The authors suggest that their experiments show something about endoderm formation in vivo without addressing this point which substantially diminishes from the interest of the manuscript.

2. The authors suggest that this view of endoderm differentiation, which doesn't require full EMT is novel, however, much of the observations here are already known. It is known that future endoderm cells do not down regulate E-Cadherin but instead must continue to express it. They also are known to migrate collectively rather than as single cells in a cadherin-dependent way (Montero et al., Development 2005; reviewed in Nowotschin et al., Development 2019). The authors should discuss this literature and make clear which aspects of the proposed mechanisms are novel.

3. The authors are assessing the status of EMT based on a single marker, E-Cad. If this is a major point of the manuscript other markers e.g. Snail, N-Cad should be examined.

4. It is well known that embryoid bodies form an outer layer of visceral endoderm e.g. Concouvanis and Martin Cell 1995, Doughton et al., PLOS One 2010. None of the markers here are exclusive to definitive endoderm (including Sox17 which is used throughout, see Artus et al., Dev Biol 2011). The authors should address the possibility that their observations may be consistent with a similar mechanism and may not reflect definitive endoderm differentiation.

[Editors’ note: further revisions were suggested prior to acceptance, as described below.]

Thank you for resubmitting your work entitled "Cell-state transitions and collective cell movement generate an endoderm-like region in gastruloids" for further consideration by *eLife*. Your revised article has been evaluated by Didier Stainier (Senior Editor) and a Reviewing Editor.

The manuscript has been improved but there are some remaining issues that need to be addressed, as outlined below:

Conceptual concern:

The notion that endoderm specification does not requite EMT is indeed backed up by data in the embryo, and this part is now adequately covered in the paper.

The observation that, in gastruloids, endoderm cells arise as islets that get sorted from the other cells possibly though differences in cell junctions tension is interesting and well substantiated. It is supported by studies in vitro in different species, including rather ancient experiments. However, the referees point that the evidence that a similar mechanism is at play in the mouse embryo is still weak, and would suggest tempering that claim (revise paragraph starting at line 233 in discussion).

Technical concern:

Although the reviewers acknowledge that the authors have increased the number of gastruloids and cells in most quantifications, they argue that the number is not huge yet in view of the variability. Therefore they propose that:

– number of samples and replicates should be added to the figure legends for clarity.

– statistical analysis should be reinforced through addition of a power calculation.

Format:

Reviewers point that the paper would be better suited as a short report. A change of format at this stage may present a risk of losing clarity. We would nonetheless recommend to condense the data notably through fusion of the supplementary figures in one (or maximum two) per main figure.

Please find below the comments of an additional reviewer consulted.

*Reviewer #4:*

Hashmi et al., present a gastruloid model for the emergence of endoderm-like cells after treatment with a Wnt agonist. These gastruloids appear to resemble the posterior region of the mouse embryo during gastrulation, segregating into T+ and Foxa2+ regions, the latter of which retains E-cadherin, paralleling recent studies which have suggested endoderm cells do not undergo EMT on exit from the epiblast. The data they present are intriguing, and suggest a conserved mechanism for sorting mesoderm and endoderm, however it is very difficult to draw direct comparisons to the mouse embryo. More information on the emergence, behaviour and ultimate fate of these endoderm-like cells would better substantiate their model as well as provide valuable new information, however this is technically challenging and would result in a much larger study than what is presented here. I agree with Reviewer 1 that the fit for this manuscript is more suited for a short report than a full research article.

In general, the authors have addressed each reviewer's points, provided new data as well as clarified their methods and increased the number of gastruloids used in each analysis. While the number of samples and replicates used are presented in a supplementary table, it would be helpful for the reader if these were included either in the figure legends or the figure themselves. The increase in sample number is not huge, however, (particularly given the variability between gastruloids) and the study could benefit from some power calculations. While there are clearly more avenues for the authors to expand up on or follow up, regarding the manuscript in its current state they have addressed the majority of the reviewer's questions and concerns.

---

## [Author Response]

Reviewer #1:General assessmentThe manuscript by Hashmi et al., describes the emergence of endoderm-like cells in a stem cells based embryo model. The particularity of the protocol is that it stimulates transition through an epiblast-like state, then differentiation towards mesoderm after a pulse of Chiron, a Wnt agonist. In those conditions, islets of E-cadherin positive cells emerge, surrounded by Ecad+Brachyury+, then Brachyury positive cells. Those islets fuse together at the tip, possibly due to distinct surface tension and directed cell movements, and express endoderm markers such as Sox17 and FoxA2.It is an original approach and concept, raising new questions and possibilities about the mechanisms of endoderm emergence in the mouse embryo. The manuscript is well written and clear.

We thank the reviewer for the positive comments.

Concerns1.The data would benefit from increased clarity in stating, for each experiment, the proportion of aggregates in which a given phenomenon was observed, as well as the number of cells counted in each aggregate, in particular in supplementary figures.

We have increased the number of experiments and are providing in a table (Table 1) for each figure the number of observations (number of gastruloids, number of independent experiments).

2. For the migration analysis, it could be interesting to distinguish each cell trajectory in order to distinguish behaviours of the subpopulations.

Distinguishing each cell trajectory in a 3D free-floating aggregate is a challenging task, and we have to improve imaging and computational approaches to do so, which is beyond the scope of this manuscript. However, to address this point, we have extended the optical flow analysis on 2-photon and epifluorescence videos. We see collective cell flows toward the posterior pole and cells that recirculate backwards at the periphery of the aggregate. Analysis of cell flows and E-cad expression show a clear correlation between the two (New Figure 4 A and Video 16). Cells that recirculate backwards are Bra+ E-cad- that loose Bra expression, while cells remaining at the pole are either Bra+ or Bra-/E-cad+ (Figure 4 A-B Videos 15-16).

3. In terms of the surface tension analysis, performing a similar analysis at different timepoints might be helpful to understand how the islets come to fuse at the tip.

We have made additional analysis from confocal/2-photon images at 72 and 96h, to capture the segregation of the islets. We measured contact angles, and thus inferred ratio of tensions for E-cad expressing islands. The surface tension at the border of E-cad islands remains high but constant. The data are shown in Figure 4C, Figure 4 —figure supplement 2. The Videos of E-cad (Videos 8-11) highlight that the cells are very dynamic, dividing and moving in 3D. The high surface tension is likely to prevent the islets to disrupt and remain compact while they segregate at the tip.

4. I am not sure about the specificity of the gata6 staining, not that it adds a lot to the story.

We decided to not include this data on this staining as it is not relevant to the main points of our story.

5. The authors might want to discuss how those aggregates evolve, and whether the endoderm-like cells have a potential for further differentiation.

We show that cells express Sox17 at 120 hrs-pp (Figure 2 —figure supplement 2). A recent manuscript from Lutolf lab (Vianello and Lutolf, biorXiv 2021) reports that different endodermal types arise within 7 days of culture, including expression territories corresponding to anterior foregut, midgut and hindgut. Our study focuses on the symmetry breaking events leading to the early endoderm formation, prior to this specification. Vianello and Lutolf was not submitted at the time of the 1st submission of our paper; we now refer to it and to its findings in the discussion.

ConclusionOverall it is an interesting and original observation, well substantiated. More details on the quantification methods would help convince about the solidity of the model: chance of obtaining those cells, amount of cells of each subpopulation including those described in supplementary figures, technical possibility of sorting them for transcriptome analysis etc. In terms of format, I think this report would better fit as a short report than full research article.

We thank the reviewer again for recognizing the originality of our work and we hope that the new data provided address their points.

Reviewer #2:In this manuscript, the authors proposed a new mechanism of endoderm formation in 3D gastruloid models based on cell migration and fragmentation. Specifically, they found that E-cad is first uniformly expressed inside mESC aggregates. After exposure to Wnt agonist Chiron (Chi), a gradual repression of E-cad and an increase of T-Bra were detected. Cells in the core are tightly packed and express E-cad. T-Bra expressing cells are sparsely wrapped around the core. A directed flow of E-cad expressing cell islands surrounded by T-Bra expressing cells help to accumulate E-cad expressing cells to the tip of the aggregate and form endoderm domain.I think the dynamical expression of E-cad and T-Bra and the directed cell flow reported in this manuscript are interesting. The results and videos have showed that the elongation and formation of endoderm region is a collective cell behavior rather than single cells undergo epithelial-to-mensenchymal transition. But I am not convinced that the process is done based on the three-step mechanism proposed by the authors. Moreover, I am not sure if this phenomenon really happened in mouse embryo development, giving the considerable differences between gastruloid model and embryo. Since there are methods culturing mouse embryo in vitro up to the early organogenesis stage, I would suggest the authors to provide more evidence showing that the proposed mechanisms might also happen in vivo.In addition, the manuscript provides too little information to understand the phenomenon. And they did not clearly introduce experimental and computational methods they used to acquire the results. I listed some of my comments below.

We are aware that our manuscript does not provide direct evidence that the mechanism that we show in gastruloids is present in the embryo. However, our data are not at odds with the existing literature, in particular two recently published works (Probst…and Arnold, Development 2020 and Schneiber et al., Nature Cell Bio 2021). The latter (which was not published at the time of our 1st submission) reports that endoderm progenitors cells do not undergo a mesenchymal transition in the primitive streak of the mouse embryo, which is consistent with our observations. As discussed in the general comments above, we now highlight the links between the scattered observations that exist in the literature. We also mention the possibility that the mechanogenetic trajectory followed by cells in the gastruloids may be different from that in the embryo. In our opinion, this does not diminish the interest of our manuscript as it reports a mechanistic description of the symmetry breaking event.

1. Did all 3D aggregates become elongated shape in the presence of Chi? If not, what do E-cad and T-Bra expressions and cell migration dynamics look like in those spherical aggregates? Without Chi, inside the spherical aggregates, do they also have cell migration since the aggregates keep growing larger?

We now provide more data of 3D aggregates in the presence and absence of Chi (Video 2). A large fraction of aggregates are already elongated in the presence of Chi at 96 hours (30/64). All the elongated aggregates at 96 hours show a single pole of T-Bra. Some aggregates are multipolar both in shape and T-Bra some (18/64). Some aggregates are not elongated at 96hrs (15/64). Such aggregates show a non-polarized pattern of E-cad (Figure 4 —figure supplement 2). In the absence of Chi, the aggregates grow isotropically as shown by the (isotropic) flow pattern (Figure 4 —figure supplement 1).

2. When did the collective cell migration start? Right after exposing to Chi? Or after some percentage of cells become T-Bra positive cells? Did the gastruloid keep elongating with directed cell flow inside it when cultured for a long time?

The collective cell movement happens after removal of Chi (See for example Video 15). The gastruloids keep elongating with direct cell flow inside it (Figure 4 A-B and Videos 12-14).

The aggregate forms a tip while the collective motion is observed. In our protocol using FGF and Activin, the aggregates do not elongate as much as in the standard protocol. However, we have not looked at aggregates beyond 120 hrs post-plating since the larger-sized aggregates require different experimental conditions (shaker in particular, making live imaging tricky).

3. Are the collective cell migration driven by the T-Bra cells? Is it a spontaneous property of E-cad cells when the E-cad cell density exceed some critical threshold (e.g. glassy dynamics)?

The correlation between maximal shear and the localization of the interface between T/Bra+ and not T/Bra- indicate the role of T/Bra cells in the collective movement (Figure 4B and Figure 4: Figure sup 1). The islands of E-cad expressing cells are within the T/Bra+ domain and thus move together with the T/Bra+ cells. We cannot currently quantify the recirculation at the interface between E-cad+/Bra- islands and E-cad-/Bra- cells as this would require a cell line to be able to track individual cells, at least a two-colour cell line such as an E-cad/histone labelled line.

4. Does the elongation and migration dynamics depend on the concentration of Chi, size of the aggregates? I noticed the authors used different initial seeding densities.

We used the Chi concentration that is commonly used in the “gastruloid” community and didn’t investigate the concentration dependence. How the elongation and collective cell movement depend on the initial size is an interesting question but is beyond the scope of our study. We are currently limited by the number of videos we can obtain with 2-photon microscopy and couldn’t perform a systematic study on the size. We used between 50 and 200 cells as initial conditions. For these different sizes, we didn’t notice qualitative differences in the flow observed by 2-photon imaging.

5. For the elongated cell aggregates, one side of cells express E-cad. How about the other side of cells? Did they all become mesoderm-like (T-Bra+) cells?

We have addressed this point by additional immunostainings. The other side is not T/Bra and shows N-cad expression (Figure 3 —figure supplement 3) and fibroblast phenotypes with cavities and possibly reach in ECM as suggested by imaging sulforhodamine B which marks extracellular space (Videos 12-14).

6. Many results are only based on several (3 or 4) gastruloids. For example, figure 1 (d) (e), figure 2 (b), figure 3(c). And in Figure 4 (b), the authors only quantify 13 junctions, probably in the same gastruloid. Due to the heterogeneity among the gastruloids, I am not sure how repeatable the experiments are. Can those observations really reflect phenomenon happened inside the majority of gastruloids? I think the authors should provide some quantifications of the percentage of observing the reported results among a large number of gastruloids.

We are now providing more data using epifluorescence, confocal, and biphoton imaging on live and stained gastruloids. The number of gastruloids/cells and replicates is provided in Table 1.

Unclear results or experimental descriptions:1. Can the authors show a schematic of the experimental process, such as the time of adding Chi and fixation?

We added a small panel in the main Figure 1 to make this clear.

2. 'We find that 30/37 … set to 0.125.' How did the authors define and calculate the elongation ratio and E-cadherin polarization ratio? How did the authors define the elongation threshold?

The elongation is measured by fitting an ellipse to the aggregate contour and is computed as (1 – (minor-axis/major-axis)), also sometimes referred as ellipticity.

The elongation threshold was selected as 0.125 or the 75% quartile of the Chi – aggregates. E-cadherin polarization was computed as explained in the main text and the supplementary method: the signal polarization is defined as the maximum of contrast between the two-halves of the aggregate. It was calculated in two-steps: first the aggregate mask was chopped into two halves about the eigenvector of the inertia matrix, which passes through the centroid of the mask. The contrast was then calculated as (Imax-Imin)/(Imax+Imin) where Imax and Imin represent the intensities of the brighter and less bright halves, respectively. The contrast maximum was then determined by angular scanning by repeatedly rotating the cutting plane by 5°

3. Figure 1 (a): what is the y axis? 1 (d): how did the author measure the E-cad and T-Bra expression? Fixing at different time points or live imaging? If it is live imaging, is the acquisition process influenced by adding and removing Chi? 1(e) how can the authors get continuous results for polarization?

Elongation is measured as 1 – (ratio between the length of the minor axis and that of the major axis of the aggregates). It is measured from live imaging using epifluorescence. The polarization is measured from day 3 (Figure 1E) after Chi removal.

4. Figure 2 (b) Are those dots represents the nuclear position? Can the authors provide the 3D view of the whole gastruloid? (c) What information the authors are trying to get from the connectivity graph?

Owing to the constraints imposed by the number of markers to be stained, most stainings used for the analysis in Figure 2 B-D did not have nuclear staining. However, other nuclear markers such as Bra and membrane staining such as Ecad were utilized to determine the cell centroids. Automation could not be utilized at this step and identifying different cell populations needed manual intervention. The connectivity graph quantifies the closeness/sparseness of the different cell populations, and here it additionally tells whether a particular cell population has a homotypic or a heterotypic cell neighbourhood.

5. Figure 3 (a) What are those white dots in the images, also in video 6? Can the authors replace t1, t2, t3, t4 with the real time, such as 24h, 36h? (d) How did the authors calculate the intensity? How did the authors normalize the intensity? The schematic in (b) is hard to understand. What do the light and dark colors represent? How did the authors measure theta_1 and theta_2, especially in 3D situation? More quantitative information should be acquired from (a).

The white dots visible in Old Figure 3 A and now Videos 12-14 correspond to dead cells which are marked by sulforhodamine B (cross-talk from the red to the green channel).

We have replaced t1-t4 with the actual timings that were provided in the Figure caption. The normalization of the intensities is done by using the maximum intensities of the respective channels.

The authors think that the reviewer is referring here to figure 4b (now Figure 4C and Figure 4: figure supplement 2) for the angles. Angles can only be specified in a 2D context (between lines) even in a 3D configuration (surfaces). The angles were measured by drawing tangents to the contours at the vertex where E-cad/Ecad cells meet (and surrounded by a E-cad- cell). Both theta angles are subtended between the respective tangents and the imaginary line extending the E-cad/E-cad junction.

6. I am not able to identify islands of E-cad expressing cells in Figure 3 (a) and video 6.

Islands of E-cad expressing cells are visible in Figure 3D and Figure 3: figure supplement 1 (staining/confocal), but we agree that they are more difficult to identify from the live 2-photon data (old Figure 2A and old video 6). We clarify this point by providing high resolution 2-photon videos (Video 8 with circled islands and Videos 9 and 10).

Reviewer #3:In this manuscript Hashmi et al., describe the emergence of an endoderm population in a gastruloid model. They observe that endoderm cells are positive for E-Cad, likely express E-Cad continuously from an epiblast state, initially form small islands, and finally coalesce into a larger endoderm region at the pole of the gastruloid. There are several issues with this manuscript in its current form.1. No evidence is provided that there is an relationship between how endoderm forms in this gastruloid model and in vivo. In fact, endoderm is believed to derive from a restricted area in the anterior primitive steak. This is evident from the mouse imaging data of Mcdole et al., Cell 2018 as well as from more recent genetic labeling experiments (Probst et al., bioRxiv 2020). It is well known that cells of different germ layers may self-segregate and this may drive the behavior observed here downstream of heterogenous differentiation in the gastruloids, but that is not necessarily the mechanism which occurs in vivo. The authors suggest that their experiments show something about endoderm formation in vivo without addressing this point which substantially diminishes from the interest of the manuscript.

This is an important point and we refer to it in our opening statement. A gastruloid is not an embryo but a model for certain events that happen in the embryo. In this case our study highlights some aspects of gastrulation that may have been overlooked and makes a well-grounded proposal for the origin of the endoderm. Importantly, since our first submission, a report from Lickert and colleagues (Schneiber et al., Nat Cell Bio 2021) has shown that endoderm formation in the mouse doesn’t require mesenchymal transition, which strongly substantiates our findings and the relevance of our observations in vitro.

2. The authors suggest that this view of endoderm differentiation, which doesn't require full EMT is novel, however, much of the observations here are already known. It is known that future endoderm cells do not down regulate E-Cadherin but instead must continue to express it. They also are known to migrate collectively rather than as single cells in a cadherin-dependent way (Montero et al., Development 2005; reviewed in Nowotschin et al., Development 2019). The authors should discuss this literature and make clear which aspects of the proposed mechanisms are novel.

The recent publication from the Lickert group, subsequent to our submission, does make the point that our observations and discussion were, and are, novel. With regard to the Montero, indeed, our observations are relevant, more so as they highlight aspects of endoderm specification in mammals that have not been described before and which shed light not only on endoderm specification in these embryos but to its relationship to anamniotes. We have now added a paragraph in the discussion to highlight this point and added two very relevant references.

3. The authors are assessing the status of EMT based on a single marker, E-Cad. If this is a major point of the manuscript other markers e.g. Snail, N-Cad should be examined.

We now provide N-cadherin and Snail staining in Figure 3 —figure supplements 3 and 4. N-Cad expression anti-correlates with E-Cad. E-Cad+ cells show either no Snail expression or extranuclear Snail. This second population corresponds to the *sox2*^+^ compact islands that segregate at the tip and become Sox17+.

4. It is well known that embryoid bodies form an outer layer of visceral endoderm e.g. Concouvanis and Martin Cell 1995, Doughton et al., PLOS One 2010. None of the markers here are exclusive to definitive endoderm (including Sox17 which is used throughout, see Artus et al., Dev Biol 2011). The authors should address the possibility that their observations may be consistent with a similar mechanism and may not reflect definitive endoderm differentiation.

As discussed before, Gastruloids are not Embryoid bodies and have already been shown not to have any visceral endoderm see e.g PMID: 28951435 and also discussion of reviewers comments in Vianello and Lutolf, bioRxiv 2021.

[Editors' note: further revisions were suggested prior to acceptance, as described below.]

Conceptual concern:The notion that endoderm specification does not requite EMT is indeed backed up by data in the embryo, and this part is now adequately covered in the paper.The observation that, in gastruloids, endoderm cells arise as islets that get sorted from the other cells possibly though differences in cell junctions tension is interesting and well substantiated. It is supported by studies in vitro in different species, including rather ancient experiments. However, the referees point that the evidence that a similar mechanism is at play in the mouse embryo is still weak, and would suggest tempering that claim (revise paragraph starting at line 233 in discussion).

We have shown that in the gastruloids, endoderm cells arise as islets that get sorted from other cells through surface tension difference. We tempered the claim that a similar mechanism is at play in vivo. The associated paragraph now reads:

“The evidence that a similar mechanism is at play in the mouse embryo remains partial, although a recent report suggests that endodermal progenitors emerge in small islands (see Figure 3 in (Probst et al., 2021))”.

Technical concern:Although the reviewers acknowledge that the authors have increased the number of gastruloids and cells in most quantifications, they argue that the number is not huge yet in view of the variability. Therefore they propose that:– number of samples and replicates should be added to the figure legends for clarity.– statistical analysis should be reinforced through addition of a power calculation.

We are now reporting the number of samples and replicates in the figure legends and have also added power calculations (highlighted in yellow in the text).

Format:Reviewers point that the paper would be better suited as a short report. A change of format at this stage may present a risk of losing clarity. We would nonetheless recommend to condense the data notably through fusion of the supplementary figures in one (or maximum two) per main figure.

We have reduced the number of figure supplements with a maximum of 2 figure supplements per main figure: 1 figure supplement for Figure 1, 2 figure supplements for Figure 2; 2 figure supplements for Figure 3; 2 figure supplements for Figure 4.